# Moderately Inducing Autophagy Reduces Tertiary Brain Injury after Perinatal Hypoxia-Ischemia

**DOI:** 10.3390/cells10040898

**Published:** 2021-04-14

**Authors:** Brian H. Kim, Maciej Jeziorek, Hur Dolunay Kanal, Viorica Raluca Contu, Radek Dobrowolski, Steven W. Levison

**Affiliations:** 1Department of Pharmacology, Physiology and Neuroscience, Rutgers-New Jersey Medical School, Newark, NJ 07103, USA; kimbh@njms.rutgers.edu (B.H.K.); hkanal@ptcbio.com (H.D.K.); 2Federated Department of Biological Sciences, New Jersey Institute of Technology, Rutgers University, Newark, NJ 07102, USA; maciejjeziorek94@gmail.com (M.J.); raluca.contu@rutgers.edu (V.R.C.); radek.dobrowolski@rutgers.edu (R.D.); 3Glenn Biggs Institute for Alzheimer’s & Neurodegenerative Diseases, University of Texas Health Sciences Center, San Antonio, TX 78229, USA

**Keywords:** encephalopathy, autophagy, cell death, premature birth, neuroprotection

## Abstract

Recent studies of cerebral hypoxia-ischemia (HI) have highlighted slowly progressive neurodegeneration whose mechanisms remain elusive, but if blocked, could considerably improve long-term neurological function. We previously established that the cytokine transforming growth factor (TGF)β1 is highly elevated following HI and that delivering an antagonist for TGFβ receptor activin-like kinase 5 (ALK5)—SB505124—three days after injury in a rat model of moderate pre-term HI significantly preserved the structural integrity of the thalamus and hippocampus as well as neurological functions associated with those brain structures. To elucidate the mechanism whereby ALK5 inhibition reduces cell death, we assessed levels of autophagy markers in neurons and found that SB505124 increased numbers of autophagosomes and levels of lipidated light chain 3 (LC3), a key protein known to mediate autophagy. However, those studies did not determine whether (1) SB was acting directly on the CNS and (2) whether directly inducing autophagy could decrease cell death and improve outcome. Here we show that administering an ALK5 antagonist three days after HI reduced actively apoptotic cells by ~90% when assessed one week after injury. Ex vivo studies using the lysosomal inhibitor chloroquine confirmed that SB505124 enhanced autophagy flux in the injured hemisphere, with a significant accumulation of the autophagic proteins LC3 and p62 in SB505124 + chloroquine treated brain slices. We independently activated autophagy using the stimulatory peptide Tat-Beclin1 to determine if enhanced autophagy is directly responsible for improved outcomes. Administering Tat-Beclin1 starting three days after injury preserved the structural integrity of the hippocampus and thalamus with improved sensorimotor function. These data support the conclusion that intervening at this phase of injury represents a window of opportunity where stimulating autophagy is beneficial.

## 1. Introduction

Neonatal encephalopathy is a common cause of neurological morbidity in infants, occurring in 3 per 1000 live births annually. While the etiology of encephalopathy can be nonspecific and heterogeneous, hypoxia-ischemia (HI) remains the predominant cause of neurologic impairment in 50% of all cases [1]. As the name implies, HI injury arises from inadequate oxygenation/perfusion to the fetus during birth (e.g., asphyxiation) [2]. Clinicians currently use a combination of maternal medical history and physical exam findings to diagnose HI encephalopathy, as more specific and reliable markers of injury have not been identified [3]. While neuroimaging can evaluate the pattern and severity of injury, it may not be included in the initial clinical assessment. Such shortcomings increase the time in which an acute HI event is accurately determined, compromising successful interventions.

Clinical and experimental data continue to bolster the view-point that encephalopathy due to HI is an “evolving process” in which the initial injurious event triggers a cascade of death effectors in the weeks that follow [4]. Diffusion-weighted brain imaging studies of HI neonates taken across three to four days capture an expansion of the initial superficial lesion of the neocortex to involve deeper regions of the brain [5]. Within the first 6 h of injury, cells within the core of the infarction undergo necrosis, having received the brunt of ischemic injury [6]. After this acute stage of primary cell death, surviving cells within the penumbra face stored energy depletion, mitochondrial dysfunction, ion transport failure and accumulation of free radicals and excitatory amino acids [7,8]. Cellular death is predominantly apoptotic during this phase, with caspase-3 and apoptisis inducing factor-AIF dependent pathways steadily increasing up to 48 h after injury [9]. We and others have identified this stage as a period of secondary energy failure and neurodegeneration [10,11,12]. Beyond 72 h after injury, the damaged brain can continue to deteriorate as inflammation persists [10,13]. In this tertiary stage, persistent maladaptive glial activation can sustain a pro-inflammatory environment aggravating neuronal cell death. In moderate HI injury, apoptotic cell death in the peri-infarct region steadily increases for up to two weeks post-injury, eventually surpassing death seen at 24 h [9]. Given this time window, these cells represent an important target for therapeutic interventions.

We previously showed that inhibiting the TGFβ type I receptor activin-like kinase 5 (ALK5) using the small molecule antagonist SB505124 significantly improved neurological outcome, even when administered as late as three days after injury in a rat model of late pre-term HI [14]. Hemispheric volume measurements indicated that two brain structures particularly susceptible to HI injury (i.e., hippocampus and thalamus) were preserved up to three weeks past the initial insult [14,15]. This led to preserved sensorimotor function and to improved learning and memory, indicating that ALK5 inhibition can confer long-term protection and maintain neurological integrity [15]. Thus, the tertiary stage of HI injury represents a promising window for therapeutic intervention that needs to be evaluated in clinical trials, specifically via TGFβ-signaling inhibition.

Delayed ALK5 inhibition preserved several important brain structures following HI presumably by reducing tertiary cell death, but additional studies remained to elucidate the mechanism(s) through which those structures are preserved. Here, we subjected rats on P6 to the Vannucci HI model of brain injury, which produces a unilateral brain injury to study late preterm injury [16]. We evaluated the level of cell death occurring after HI injury and investigated the effects of ALK5 inhibition on the level of autophagy, a neuroprotective process responsible for clearing cellular debris in the lysosome. Correspondingly, we evaluated the efficacy of the autophagy inducing peptide, Tat-Beclin1, to determine whether directly enhancing autophagic flux would reduce tertiary cell death after HI injury.

## 2. Materials and Methods

### 2.1. Rodents

All experiments were performed in accordance with research guidelines set forth by Rutgers New Jersey Medical School IACUC and were in accordance with the National Institute of Health Guide for the Care and Use of Laboratory Animals (NIH Publications No. 80-23) revised in 1996 and the ARRIVE guidelines. Timed pregnant Wistar rats at embryonic day 18 of gestation were purchased from Charles River Laboratories (Wilmington, MA, USA). Following delivery, litter sizes were adjusted to 12 pups per litter, and efforts were made to ensure the number of each sex and pup weights were equal and consistent. Animals were group housed and kept on a 12-h light/dark cycle with ad libitum access to food and autoclaved water. Rat pups remained undisturbed with the dam until the day of HI injury. Following injury, the pups were returned to their respective dam.

### 2.2. Neonatal Hypoxic-Ischemic Brain Injury

Cerebral HI in six-day-old Wistar rat pups (P6, day of birth = P0; mean body mass = 15 g) as a model of late pre-term injury was performed as previously described [14,17]. Briefly, rats were anesthetized with isoflurane (3–4% induction, 1–2% maintenance) prior to right common carotid artery cauterization. A special effort was made to carefully isolate the carotid without damaging other structures contained within the carotid sheath (i.e., internal jugular vein and vagus nerve). The neck incision was sutured with 4-0 surgical silk. Following a one-hour recovery period, rats were exposed to 75 min of hypoxia in humidified 8% oxygen/nitrogen balance. Sham rats were anesthetized and underwent isolation of the right common carotid without cauterization and then were exposed to hypoxia. Rats in each litter were randomly assigned to experimental groups after HI injury. Sample sizes per experiment were chosen to achieve sufficient statistical power with minimal numbers of animals based on pilot studies.

### 2.3. SB505124 Drug Delivery

Three days following HI injury, rats were anesthetized (isoflurane, 3–4% induction, 1–2% maintenance) and an incision was made in the subcapsular region. Osmotic pumps (Alzet 1007D; Durect, Cupertino, CA, USA) were loaded with either vehicle (sodium citrate buffer with 30% DMSO *v/v*) or 30 mM of the ALK5 pharmacological inhibitor, 2-(5-Benzo [1,3] dioxol-5-yl-2-tert-butyl-3HImidazol-4-yl)-6-methylpyridine hydrochloride hydrate (SB505124) (Sigma-Aldrich; St. Louis, MO, USA) and implanted subdermally. The incision was then sutured with 4-0 surgical silk, and animals were surveyed twice per day for signs of infection and/or distress. The 1007D model osmotic pump, which can continuously deliver solutions for seven days, was left implanted until euthanasia. Animals subjected to analyses were treated with SB505124 for four days, following the same dosing schedule as in our previous studies [14].

### 2.4. Tat-Beclin1 Administration

Three days following HI injury, rats were injected intraperitoneally (i.p.) with 50 uL Tat-Beclin1 (Cat # #506048, EMD Millipore; Billerica, MA or Cat #S8595 Selleck Chemical, Houston, TX, USA) at 15 mg/kg or vehicle (PBS). To assess brain penetrance, brain samples were collected 48 h after injection and processed for Western blot analysis and immunostaining of autophagic markers LC3 and p62. For sensorimotor function assessment at P20, rats were injected with Tat-Beclin1 or vehicle once on Day 3 and again on Day 5 after injury.

### 2.5. Organotypic Slice Culture

Three days following HI, rats were deeply anesthetized, rapidly decapitated and the brains removed. Brains were placed onto a steel brain matrix (Stoelting Co.; Wood Dale, IL, USA) immersed in cold DMEM media, and 1000 μm-thick coronal slices were cut using a microtome blade spanning the region that was within the territory affected by injury (approx. Bregma +1.0 to −3.0 mm) [16]. Cut slices were transferred to a six-well tissue culture cluster with sterile DMEM supplemented with 20% horse serum (*v/v*). Once plated the media was exchanged with fresh DMEM w/20% serum (control) or DMEM w/20% serum +15 μM SB505124 and the slices were placed into an incubator at 37 °C, 5% CO_2_. Particular effort was made to ensure that the slices were never fully submerged in the media to ensure proper gas exchange during incubation. After 1 h, the media were exchanged with fresh DMEM w/20% serum or DMEM w/20% serum +200 μM chloroquine for lysosomal inhibition. After an additional hour, the slices were washed with cold DMEM, and the hemisphere ipsilateral to HI injury was collected in cold lysis buffer for Western blot.

### 2.6. Sensorimotor Tests

Two weeks following HI injury, rats were subjected to a battery of sensorimotor function tests. Tests were conducted by an investigator blinded to the experimental groups.

Modified Neurological Severity Test (mNSS): The mNSS is comprised of a series of 11 different tests that are evaluated and aggregated into a modified neurological severity score as described in detail previously [14].

Beam walking tasks: Rats were tested on horizontal beam and inclined beam walking tasks. For the horizontal beam the rats were placed at the end of a 2.5 cm-wide, 80 cm long wooden beam that was suspended 42 cm above the ground. A dark box with bedding was at the other end of the beam and served as a target for the rats to reach. For the inclined beam walking test, an elevated (80 cm in length and 2 cm in width) wooden beam was placed at a 30° angle. The number of foot slips (either hind legs or front legs) and the time to traverse each beam was recorded and assessed. Decreased performance on an inclined beam has been linked with decreased subcortical white matter integrity after injury. Failure to climb the beam in less than 15 seconds was considered a failure.

Hang test: The rats were allowed to grasp a 1 cm diameter bar with their forelimbs and the time that the rat held onto the bar was measured. Each rat was tested over three nonconsecutive trials.

### 2.7. Western Blot Analyses

Microdissected brain tissue from the injured (ipsilateral) and uninjured (contralateral) hemispheres was collected. The tissue was homogenized and then sonicated in lysis buffer. Thirty micrograms of denatured protein were loaded onto a 4–12% Bis-Tris gel (Invitrogen, Carlsbad, CA, USA) and 5 μL of Amersham ECL Rainbow Marker was loaded as a molecular weight standard (GE Life Sciences, Pittsburgh, PA, USA). Proteins were transferred onto nitrocellulose and incubated with primary antibody: LC3 (rabbit polyclonal, Cell Signaling, cat # 12741S, 1:1000), SQSTM1/p62 (guinea pig polyclonal, American Research Products, Cat # 03-GP62-C, 1:1000), Actin (mouse monoclonal, Sigma-Aldrich, cat#A5441, 1:1000). Membranes probed for LC3 and Actin were washed with 0.01% TBS-Triton X, incubated in HRP-conjugated secondary antibodies (donkey anti mouse HRP, Jackson ImmunoResearch, Cat. #715-035-150 or Goat anti rabbit HRP: Cell Signaling Technology, Cat. # 7074S). Membranes probed for p62 were washed with 0.01% TBS-Triton X and incubated in biotinylated anti-guinea pig (goat polyclonal, EMD Millipore, cat #AP193B, 1:2500) secondary antibody, and later in Streptavidin-HRP (Thermo-Fisher, Cat # 21126, 1:2500). Membranes were washed, and bands visualized using Western Lightning chemiluminescence reagent (PerkinElmer, Wellesley, MA, USA). Imaging was performed using a BioRad ChemiDoc Imaging System combined with Image Lab software (Hercules, CA, USA).

### 2.8. Brain Histology and Immunofluorescence

In situ end labeling (ISEL): One week after HI injury, rats were deeply anesthetized with sodium pentobarbital before intracardiac perfusion with 4% paraformaldehyde (PFA) in PBS. Brains were post-fixed overnight in 4% PFA/PBS, cryoprotected with 30% sucrose overnight and embedded in Tissue-Tek OCT matrix (Sakura Finetek, Torrance, CA, USA). Serial coronal sections of 25–30 μm thickness were taken through the hippocampal and thalamic regions using the cryostat at −14 °C and mounted on slides. Sections were dehydrated and rehydrated in ethanol and water and incubated with 10 μM dNTP mix containing Digoxigenin-dUTP and 20 U/mL Klenow Fragment (Roche; Basel, Switzerland) at room temperature for 2 h. DIG-labeled nucleotides were detected using Anti-Digoxigenin-Fluorescein (sheep polyclonal, Roche, 1:75) incubated overnight at 4 °C. Images were collected by an investigator blinded to each group using an Olympus Provis fluorescent microscope. Images were captured using a Q-imaging mono 12-bit camera interfaced with iVision 4 scientific imaging software (Scanalytics, Rockville, MD, USA). Signal intensities were quantified using Fiji with plugins.

Autophagy immunofluorescence: Frozen coronal sections of 25–30 μm thickness were taken through the hippocampal and thalamic regions using a cryostat at −14 °C and mounted on slides. Sections were then incubated with primary antibodies in 1% goat serum/0.05% Triton X-100/PBS at 4 °C overnight. Primary antibodies included: (1) guinea pig anti-p62 (American Research Products, 1:300); (2) rabbit anti-LC3 (Cell Signaling, 1:200); and (3) mouse anti-NeuN (mouse monoclonal, Millipore, cat # MAB377, 1:100); and mouse anti-S100β (Sigma S2532, 1:500). Sections were washed with 0.05% Triton X-100/PBS three times for 30 min and incubated with secondary antibodies for 2 h at room temperature. Secondary antibodies included: donkey anti-guinea pig Cy5; donkey anti-rabbit Alexa 488; and donkey anti-mouse Cy3 (all from Jackson ImmunoResearch, 1:250). Sections were washed with 1% goat serum/0.05% Triton X-100/PBS three times and mounted in Prolong Gold Antifade Mount with DAPI (Thermo Scientific, Waltham, MA, USA). Confocal images were collected by an investigator blinded to each group using a Zeiss spinning-disc microscope and ZEN software. All acquired images used the same acquisition and laser settings, set initially using Sham (uninjured) samples. Images were processed such that p62 signals (detected in the far-red region) were converted to red and NeuN signals (detected in the red region) were converted to blue for signal colocalization studies. Signal intensities were quantified using Fiji with plugins [18].

LC3-p62 colocalization analyses: Three images were captured from each brain (n = 3 animals per group) for a total of 45–60 cells per determination in each brain region (neocortex, white matter, hippocampus and thalamus). The images were coded to blind the investigator to group identities and Auto-threshold (provided in Fiji plugin bundle) was performed to eliminate potential bias during elimination of background signal. Manders’ colocalization coefficient (MCC) was determined using the JACoP plugin for Fiji. The M1 value reported represents the fractional overlap of p62 signal in compartments containing LC3 signal. A value of 1 represents complete overlap of both signals; a value of zero represents no overlap. The benefits of reporting the MCC over Pearson’s colocalization coefficient or Manders’ overlap coefficient is reviewed by Dunn et al. (2011) [19].

Structural analysis: Two weeks after HI injury in Tat-Beclin1 treated rats, whole brains were extracted and dehydrated in 70% ethanol and then embedded in paraffin. Brain sections were cresyl violet stained and imaged using an Olympus SZXY brightfield microscope with CCD camera and acquired on PictureFrame software (Optronics; Goleta, CA, USA). The regions of ipsilateral/contralateral hippocampus and thalamus were labeled using the polygonal trace tool and area was determined via ImageJ. Six coronal sections were analyzed per animal, and the average percentage of each structure was compared to its corresponding contralateral side.

### 2.9. Data Analyses and Statistics

Raw data from image analyses and behavioral tests were imported into Prism (GraphPad Software; La Jolla, CA, USA) for statistical analyses using either one-way or two-way ANOVA followed by Tukey’s post hoc intergroup comparison. Graphs were produced in Prism and error bars denote standard error of means (SEMs). Significance among colocalization coefficients in immunofluorescence imaging was determined using ANOVA followed by Tukey’s post hoc intergroup comparison.

## 3. Results

### 3.1. Delayed SB505124 Reduces the Number of Apoptotic Cells in the Neocortex after HI Injury

To determine whether the tissue preservation seen with delayed SB505124 treatment was due to a decrease in the number of actively apoptotic cells, we collected brain tissue four days after SB505124 administration (equal to one week after injury) and stained sections using ISEL. This method of nicked DNA strand labeling detects cells in the early stages of apoptotic cell death and yields fewer false positives than TUNEL [20]. Based on the established stages of neurodegeneration, we suspect that the majority of ISEL+ cells at this phase—one week after injury—were apoptotic rather than necrotic or necroapoptotic. The neocortex was chosen for analysis based on our previous study which indicated that the neocortex showed the greatest change in autophagic protein LC3 and had extensive LC3 and p62 colocalization [15]. Compared to vehicle treatment, SB505124 treatment yielded a 90.64% reduction (** *p* < 0.01) in the number of ISEL+ cells in the neocortex, indicating a significant reduction in the number of apoptotic cells (Figure 1).

### 3.2. Delayed SB505124 Induces Autophagic Flux after HI Injury

Our prior data provided evidence that ALK5 inhibition altered autophagy, but the evidence was insufficient to conclude that it increased autophagic flux [15]. For example, LC3-II levels and LC3/p62 overlap can also increase when autophagosome turnover is disrupted as occurs with defects in autophagosome trafficking to lysosomes and defective membrane fusion [21,22]. Therefore, to measure autophagic flux, we prepared brain slices from the HI-injured brain three days after HI injury and then inhibited lysosomal function ex vivo using chloroquine. Levels of LC3 and p62 protein were then quantified by Western blot. To assess the effects of SB505124 on autophagic flux, slices were treated with the antagonist alone or the antagonist in combination with chloroquine.

LC3-I levels significantly declined with HI injury without any treatments added (labeled Media Only, M.O., F (3, 36) = 32.78; * *p* < 0.05); this difference disappeared in chloroquine and SB treated groups (Figure 2A,B). Incubation with SB505124 reduced LC3-I levels in Naïve (** *p* < 0.001), Sham (** *p* < 0.001), and HI groups (trending, n.s.) compared to no treatment. LC3-I significantly increased in HI brain slices treated with SB505124 + chloroquine compared to either chloroquine (^###^
*p* < 0.0001) or SB505124 alone (^$$$^
*p* < 0.0001). The increase in LC3-I obtained with the combined treatment was greater than Naïve or Sham-injured controls, but these differences were not statistically significant. Likewise, levels of the membrane-bounded LC3-II that represents autophagosomal number, declined with HI injury without any treatments (Figure 2C). Interestingly, LC3-II levels tended to increase compared to Naïve or Sham-injured controls with SB505124 treatment and with the combination treatment. However, the differences in each instance were not statistically significant (F (3, 36) = 1.027). LC3 conversion (LC3-II to LC3-I) was used as a measure of autophagosome production. This ratio largely did not change across groups and treatments, however, there was a trend for an increase in ratio in HI injury with SB505124 treatment (Figure 2D, F (3, 36) = 2.670; *p* < 0.09). Levels of p62 decreased with HI injury without treatment and with chloroquine or SB505124 treatment. With combined treatment with SB505124 and chloroquine, p62 levels rose significantly in the HI brain slices compared to Naïve controls (F (3, 36) = 2.964; ** *p* < 0.001, Figure 2E). This increase of p62 levels was significantly different when compared to HI injury with chloroquine (^##^
*p* < 0.001) or SB505124 treatment alone (^$$^
*p* < 0.001) and provides strong evidence that SB505124 treatment increases autophagic flux.

### 3.3. Independently Augmenting Autophagy after HI Injury Improves Sensorimotor Performance and Limits Long-Term Neurodegeneration

After determining that SB505124 administration increased autophagic flux in the brain, we wanted to determine whether directly stimulating autophagic flux during the same interventional period used for SB505124 would preserve brain cells and improve functional outcomes. We administered the autophagy inducing peptide, Tat-Beclin1, i.p. three days following HI injury, which is approximately the same time point at which SB505124 was administered in our previous experiments [15]. First, we determined whether the systemic delivered peptide could penetrate into the CNS to alter autophagic protein levels (Figure 3). A dosage of 15 mg/kg (administered once) was chosen as a previous study had shown its efficacy in inducing autophagy in rodents [23]. A higher dosage was decided against to prevent over-stimulation which may drive cell death via autosis [24]. By Western blot of brain homogenates collected 48 h after the Tat-Beclin1 administration, p62 levels decreased compared to shams (Figure 3A,B; F (2, 14) = 11.40; ** *p* < 0.001) and vehicle, (*p* = 0.13), and the LC3-II/LC3-1 ratio was elevated in both vehicle (F (2, 14) = 15.96; *p* < 0.005) and Tat-Beclin1-treated groups (*p* < 0.01, Figure 3C) and when vehicle was compared to Tat-Beclin1 treated rats (*p* < 0.1). Additionally, p62/LC3 immunofluorescence overlap increased in NeuN+ neurons of Tat-Beclin1 treated brains (Appendix A), whereas p62/LC3 immunofluorescence overlap did not increase in S-100b+ astrocytes (Appendix A). These changes were apparent at 48 h after Tat-Beclin1 injection, suggesting that the peptide had lasting effects on the brain at this dosage in the rat.

Given that a single dose of Tat-Beclin1 induced autophagy 48 h after injection, we administered a second injection 72 h after the first for longer timepoint assessments. This timeline was chosen to approximate the drug bioavailability of one-week SB505124 administration (Figure 4A). Two weeks after injury and Tat-Beclin1 administration, rats were subjected to behavioral analyses to assess sensorimotor deficits. The body weight of each rat was tracked during this two-week period. Vehicle-treated HI-injured rodents weighed significantly less than their Sham control counterparts (F (3, 78) = 289.4; * *p* < 0.05, Figure 4B) while there was no difference in weights in animals treated with Tat-Beclin1 versus sham controls. Measurement of the gross hemispheric volume, which was then rendered as a hemispheric ratio (injured ipsilateral area to contralateral area), revealed a significant loss of brain tissue in vehicle-treated rats (F (2, 15) = 24.83; *** *p* < 0.0001 vs. Sham, ** *p* < 0.001 vs. HI TB1), while there was no significant difference in the ratio in either the Tat-Beclin1 treated or Sham-injured animals (Figure 4C; *p* < 0.0001).

In measures of sensorimotor performance, vehicle-treated rats performed the worst, with the greatest number of foot slips per run on the horizontal 2.5 cm beam (1.48 ± 0.21 slips) and incline beam tests (1.11 ± 0.12 slips) (Figure 4D–E; F (2, 24) = 7.301; *p* = 0.0033). Tat-Beclin1 treated rats had significantly fewer foot slips than vehicle-treated rats on the horizontal beam (0.81 ± 0.22 slips, * *p* < 0.05, Figure 4D), and did not perform any differently compared to Sham controls. Rats were also assessed on the ability to hold onto a horizontal bar using their forelimbs: Sham animals held on the longest on average (14.04 ± 3.48 sec), followed by Tat-Beclin1 treated animals (12.43 ± 2.42 sec) (Figure 4F; F (2, 23) = 4.045; *p* = 0.0312). There was no statistical difference between the Tat-Beclin1 group and the Sham control. Vehicle-treated rats performed the worst on this assessment, holding onto the bar for significantly less time than Sham rats (4.06 ± 1.07 sec, * *p* < 0.05). Sensorimotor metrics were tabulated into a modified neurological severity score (mNSS) system, in which a higher score denotes poorer performance and consequently greater impairment (Figure 4G; F (2, 24) = 13.21; *p* = 0.0001). Vehicle-treated animals scored significantly higher (2.22 ± 0.32) compared to Sham animals (0.56 ± 0.18, ** *p* < 0.001) while Tat-Beclin1 treated animals exhibited some neurological impairment, but they were less impaired than the vehicle-treated rats (1.67 ± 0.17). Once again, Tat-Beclin1 treated animals showed no statistical difference from the Sham controls.

Volume assessments for the hippocampus and thalamus showed preservation of both structures with Tat-Beclin1 treatment. The hippocampus in Tat-Beclin1 treated rats retained 80.09 ± 5.27% of its size compared to the contralateral side, while the hippocampus in vehicle-treated rats was 53.81 ± 10.63% of the contralateral side (Figure 5A,B). The difference in hippocampal size was statistically significant (F (2, 20) = 12.78; ** *p* < 0.001) between Sham and vehicle-treated groups, but not statistically different between Sham and Tat-Beclin1 treated groups (*p* = 0.0003). The thalamus in Tat-Beclin1 treated rats retained 91.88 ± 3.08% of its size compared to the contralateral side, while the thalamus in vehicle-treated rats was 67.09 ±9.01% of the contralateral side, but the difference was not statistically significant (Figure 5A,B, *p* = 0.15).

A Pearson correlation coefficient was computed to assess the relationship between thalamic volume and horizontal beam performance among treatment groups. There was a strong negative correlation apparent in vehicle-treated animals (r = −0.964, F (1, 3) = 39.14; *p* = 0.008) with a similar trend in Tat-Beclin1 treated rats (r = −0.656, F (1, 3) = 2.267; *p* = 0.229). There was no correlation in the Sham group (r = 0.359, *p* = 0.129).

## 4. Discussion

Infants suffering from neonatal encephalopathy due to HI injury face lifelong neurological deficits, ranging from mild learning difficulties to severely debilitating epilepsy, cerebral palsy and cognitive disorders. As supportive management of moderate to severe encephalopathy continues to improve, the need for effective therapeutics that produce long-standing effects increases. Hence, it is absolutely vital to understand the physiological changes arising in the brain and the mechanisms underlying long-term neurodegeneration that occurs with this injury.

Many studies on perinatal HI have sought to elucidate the mechanisms of cell death towards producing therapeutics to prevent acute neurodegeneration. The majority of studies have focused on the apoptotic cell deaths that occur within the first 6–72 h—the interval of secondary cell death. These studies have shown that caspase inhibitors confer little neuroprotection when administered soon after the injury [11,25]. Other studies have shown that neurons undergo apoptosis in caspase-3 deficient mice [26] and also undergo caspase-independent forms of cell death following HI [24,27]. Evidently, more novel strategies that interrupt HI induced cell death occurring in the secondary and tertiary stages of injury must be developed.

We have previously shown that neuroinflammation and subsequent brain damage can be attenuated with systemic SB505124 administration. However, the mechanisms responsible for the reduced progression of neuronal death with SB505124 treatment was unknown [14,15,17]. Our findings now indicate that SB505124 treatment sharply reduces apoptotic cell death in the neocortex one week following injury. When coupled with our prior volumetric studies, these data suggest that TGFβ type 1 receptor inhibition reduces the number of dying cells to preserve brain volume.

### 4.1. Investigating Autophagic Flux Ex Vivo

Western blot of the injured tissue following one week of SB505124 administration showed significantly elevated LC3-I which could be due to increased synthesis of LC3-I or evidence of impaired autophagosome clearance [15]. Our slice culture data provided evidence that autophagic flux was increased when SB505124 was given three days after injury. Interestingly, previous studies have shown that autophagy is enhanced after HI but becomes progressively inhibited, coinciding with an increase in neuronal death [28]. Lechpammer et al. (2016) showed that downstream mTORC1 targets such as p70/S6 kinase and 40S ribosomal protein S6 were activated following HI injury in P6 rats and multiple studies have shown that increased mTOR activity inhibits autophagy [29]. An analysis of post-mortem human brain tissue from asphyxiated infants showed a seven-fold increase in LC3 puncta in dying neurons of the basal ganglia compared to non-injured controls, leading to the conclusion that autophagy flux and degradation of LC3 within the lysosome was greatly impaired prior to death [30]. As described for models of other neurodegenerative conditions that include Alzheimer, Parkinson and Huntington diseases, reversing impaired autophagic responses during the late stages of injury may indeed be beneficial [31,32,33].

SB505124 and chloroquine significantly increased accumulation of LC3-I and p62 in HI-injured brains. As chloroquine inhibits lysosomal protein degradation, these data indicate that SB505124 was increasing active flux rather than inhibiting autophagy. If SB505124 inhibited autophagic flux, then there would be no change in LC3-I:p62 levels in the presence of chloroquine. As further evidence, SB505124 treatment alone increased the LC3-II/LC3-I ratio, representing increased lipidation of LC3 and autophagic flux. However, the increase in LC3-II/LCD-I ratio did not reach statistical significance (*p* < 0.09) due to the small number of animals analyzed (n = 4–5/group).

### 4.2. Inducing Autophagy with Tat-Beclin1

Tat-Beclin1, is a cell permeant protein that activates endogenous Beclin1 by competing against its negative regulator GAPR-1/GLIPR2 on the Golgi surface. It was first identified as a potential therapeutic in 2013 [23]. The peptide induces autophagy in rodents and is well tolerated when administered daily for up to two weeks [23]. In that prior study, Tat-Beclin1 was administered i.p. at 15 mg/kg per day to mice infected with the West Nile virus where it significantly reduced brain viral titers and mortality six days after the start of dosing [23]. Use of the peptide also has been shown to promote axonal regeneration following spinal cord injury in mice [34], reduce neurotoxicity due to hyperammonemia [35] and to improve long-term memory when directly injected into the hippocampus [36]. Interestingly, in an adult model of stroke using middle cerebral artery occlusion, a small dose of Tat-Beclin1 (1.5 mg/kg) given i.p. at 6 and 13 days after injury *worsened* the neurological deficit and *increased* infarct volumes [37]. Therefore, it was not clear a priori that administering Tat-Beclin1 in this neonatal model of HI injury would be beneficial.

We initiated Tat-beclin1 treatment on the same day that SB505124 treatment had begun in our previous studies to enable comparisons based on earlier studies [23]. We used a dose of 15 mg/kg, i.p. which was 10× the dose administered by Hongyun et al. (2017), and decided not to use a higher dosage to prevent over-stimulation which may drive cell death via autosis [24]. Curiously, a significant difference in mean body weight between Sham animals and vehicle-treated animals was seen starting at two weeks post-injury here and in a previous study [14]. As with SB505124 treatment, Tat-beclin1 increased the mean body weights of the HI rats. Furthermore, both hippocampal and thalamic integrity were preserved with Tat-Beclin1 treatment; these two structures also were significantly preserved in SB505124 treated animals [15]. Importantly, salvaging the thalamus correlated with the improvements in sensorimotor function.

Overall, our work supports the conclusion that the neuroprotective effects of SB505124 administration can be attributed to enhanced autophagy that reduces the incidence of apoptotic cell death that occurs several days to weeks after the injury has occurred. Our novel findings have established the basis for future studies to validate SB505124 or drugs with similar mechanisms of action as therapeutics. Our studies also showed that inducing autophagic flux by administering Tat-Beclin1 during the tertiary phase of HI injury improved both histopathological and functional outcomes. Unlike other possible neuroprotectants, both SB505124 and Tat-Beclin1 can be delivered peripherally as they penetrate into the CNS. This mode of delivery makes these treatments highly translatable into the clinic and make our studies highly promising for the treatment of moderate HI in human infants.

## Figures and Tables

**Figure 1 cells-10-00898-f001:**
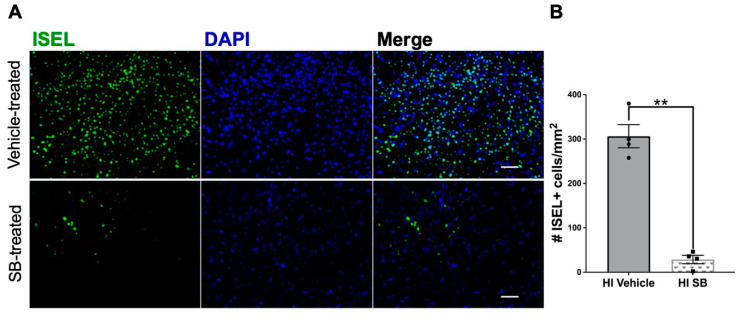
SB505124 diminished the number of actively apoptotic cells in the neocortex after hypoxia-ischemia (HI) injury. One week after HI injury at P6, samples of the injured forebrain were analyzed for actively dying cells using in situ end labeling (ISEL). SB505124 or vehicle was administered via osmotic pump beginning at three days after injury and maintained for four days prior to intracardiac perfusion. Thirty µm sections were processed for ISEL. Cells with green nuclei indicate nicked DNA strands. (**A**) Panels depict representative neocortical cells in the ischemic penumbra of injury in vehicle-treated and SB505124-treated animals. Scale bars in merged image represent 20 µm. (**B**) Quantitative analysis of the number of ISEL+ cells of the neocortex per mm^2^. n = 4 per group. Data are presented as means ± SEM ** *p* < 0.001 by Student’s *t*-test.

**Figure 2 cells-10-00898-f002:**
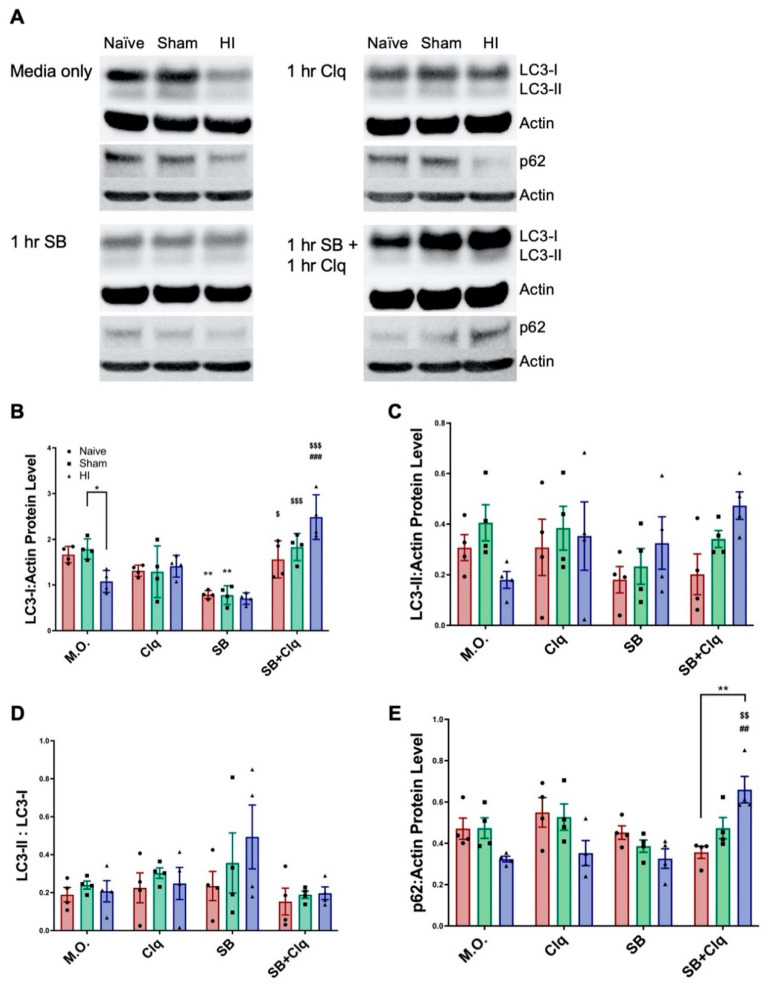
SB505124 induced autophagic flux in the injured hemisphere after HI Injury. Three days after HI, 1000 µm thick coronal slices were incubated with SB505124 (SB) or SB505124 + chloroquine (SB+Clq) in DMEM w/20% horse serum (Media Only control, M.O.) for 2 h total in 37 °C, 5% CO_2_ to assess the effect of SB505124 on active autophagy. The injured hemisphere was collected and protein extracted for Western blot. (**A**) Representative blot for LC3, p62, and β-Actin (loading control) from the injured hemisphere. (**B**) Quantitative analysis of band optical densities for LC3-I, (**C**) LC3-II, (**D**) ratio of LC3-II/LC3-I band densities, and (**E**) band optical densities for p62. n = 4–5 per group. Data are presented as means ± SEM; * *p* < 0.05, ** *p* < 0.001 when denoted by bracket; ** *p* < 0.001 for M.O. vs. SB treated slices; ^##^
*p* < 0.001, ^###^
*p* < 0.0001 for Clq treated vs. SB+Clq treated slices; ^$^
*p* < 0.05, ^$$^
*p* < 0.001, ^$$$^
*p* < 0.0001 for SB treated vs. SB+Clq treated slices by two-way ANOVA followed by Tukey’s multiple comparison test.

**Figure 3 cells-10-00898-f003:**
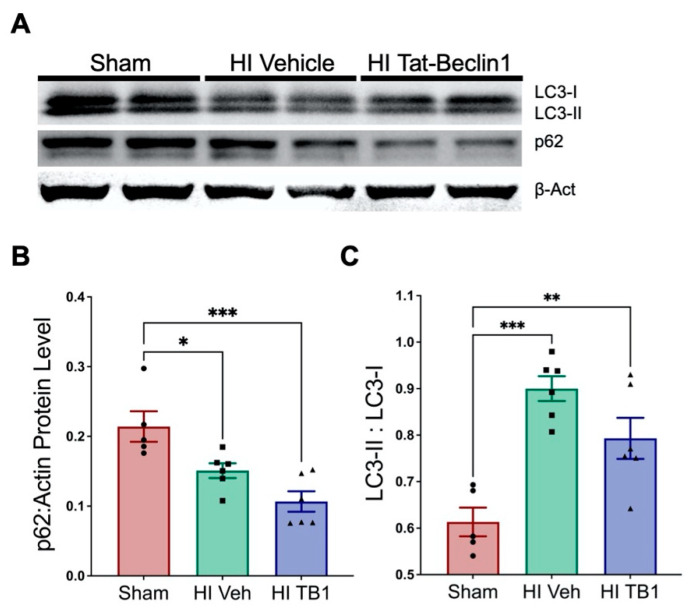
Systemic Tat-Beclin1 administration induced autophagy in the brain up to 48 h after injection. Three days after HI, rat pups were injected i.p. with Tat-Beclin1 (15 mg/kg, TB1) or vehicle (PBS). 48 h after injection, samples of the injured forebrain were prepared for Western blot. (**A**) Representative blot for LC3, p62, and β-Actin (loading control) from the injured hemisphere. (**B**) Quantitative analysis of band optical densities for p62 and (**C**) ratio of LC3-II/LC3-I band densities (Sham n = 5/group; Treatment n = 6/group). Data are presented as means ± SEM; * *p* < 0.05, ** *p* < 0.001 as determined by one-way ANOVA followed by Tukey’s multiple comparison test.

**Figure 4 cells-10-00898-f004:**
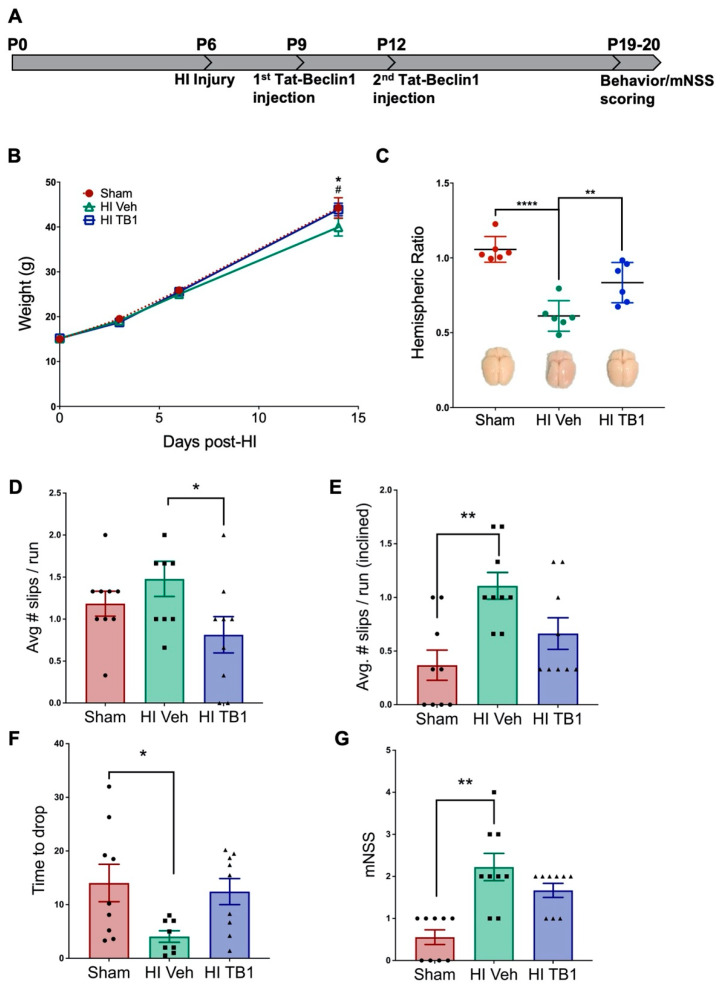
Systemically administered Tat-Beclin1 improved sensorimotor outcomes after HI injury. Two weeks after HI injury at P6, behavioral tests were performed to assess sensorimotor function. (**A**) Outline of the experimental paradigm of HI injury and behavioral testing. Three days after injury on P6, rat pups were injected once with Tat-Beclin1 (15 mg/kg), and again 72 h after the first injection. Sensorimotor testing began 13 days following HI injury. Rats were given a pre-training session 24 h before the start of testing. (**B**) Body mass of Sham-injured, HI-injured with vehicle and Tat-Beclin1 (TB1) administered rat pups tracked for two weeks after HI injury on P6. (**C**) Hemispheric ratio (IL:CL) with representative images of brains following extraction. (**D**) Average slips per run on 2.5 cm-wide balance beam. (**E**) Average slips per run on inclined 2.5 cm-wide balance beam, (**F**) Time to drop (s) for hanging bar, (**G**) mNSS score. n = 9 per group. Data are presented as means ± SEM; * *p* < 0.05, ** *p* < 0.001, *** *p* < 0.0001; ^#^
*p* < 0.05 for vehicle-treated vs. Tat-Beclin1 treated rats using one-way ANOVA followed by Tukey’s multiple comparison test.

**Figure 5 cells-10-00898-f005:**
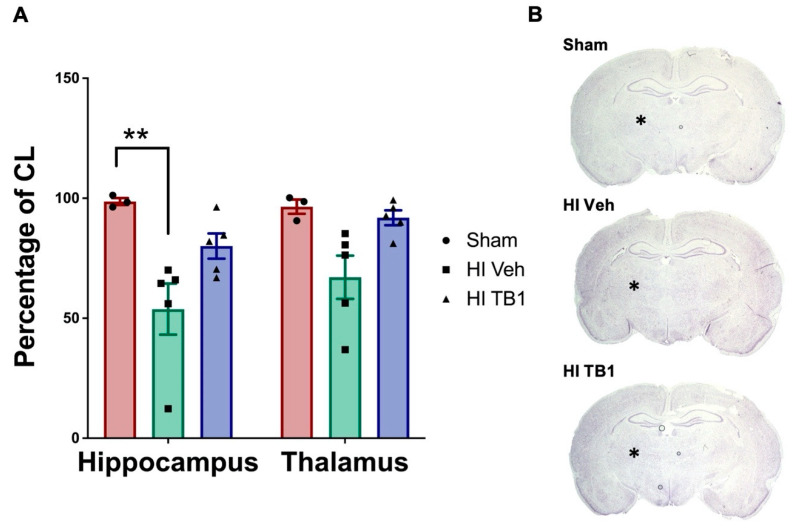
Systemically administered Tat-Beclin1 reduced hippocampal and thalamic neurodegeneration after HI injury. Cresyl Violet stained sections at +3 mm from Bregma were analyzed two weeks after HI injury at P6. (**A**) Areas of the hippocampus and thalamus were measured and normalized to contralateral structures. (**B**) Representative images of structural loss of hippocampal (enclosed by dashed line) and thalamic regions as compared to the contralateral hemisphere n = 3 for Sham group; n = 5 per HI groups. Data are presented as means ± SEM, ** *p* < 0.001 by one-way ANOVA followed by Tukey’s multiple comparison test.

## Data Availability

All data will be made available for inspection or use upon receipt of a request to the corresponding author.

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
