# Peer review of "Moderately Inducing Autophagy Reduces Tertiary Brain Injury after Perinatal Hypoxia-Ischemia"

_cells, 2021, doi:10.3390/cells10040898_

Round 1

Reviewer 1 Report

Dear editor,

We have read with great interest the article. We consider that the work is well designed and meritorious deserving publication. However, we think that the Methods section could be improved if  more information on the number of experimental subjects that have been used in each assay is included. Perhaps a flow diagram or a scheme with the number of subjects in each arm could be a good option. Also, the authors shold have included the limitations of the study

Author Response

  1. We have read with great interest the article. We consider that the work is well designed and meritorious deserving publication. However, we think that the Methods section could be improved if  more information on the number of experimental subjects that have been used in each assay is included.

Response: We have provided the numbers of subjects used for each experiment in the figure legends as we feel that it is more important to provide this information where it will be immediately accessible rather than buried in the Methods.

  1. The authors should have included the limitations of the study

Response: We have added some limitations of the study to the Discussion and also softened some of the conclusions.

Reviewer 2 Report

The study by Kim et al examined thoroughly the role of autophagy to reduce ischemia-hypoxia-induced brain injury in a late phase. The study is original and may represent a good bridge toward a potential therapy, but some points need be assessed to warrant publication.

The number of animals used, despite statistical significancy of some reported data, is not high. I understand that meeting the Three R’s and economical principles is a must in much research work, but here n does never exceed 5, with some plots with n=3. In the lack of more robust numerosity, which represents a true limitation to be highlighted, this manuscript, although interesting, should thus be labelled as “preliminary”.

The low contrast in immunofluorescence photographs are prevents visual appreciation of therein contained information, both at the PC screen and when printed on paper. Increase the contrast for the blue and red colors.

The representation of data as bar graphs mean/SEM when n is so low induces perplexity. The usage of box and whiskers or reporting individual points as in Figure 4c is a good practice for transparency to be used for all the plots. Report the ANOVA values when comparing more than two groups.

The aims are not clear as in the abstract section. If the numerosity of the animals is not increased, this remains a preliminary work and the conclusions are to be downgraded.

Minor

The introduction is quite general, especially the second para (lines 52-62) that belongs to a review rather than to a research article. The relevance of the study may be better appreciated if the Authors go directly to the central issues.

Line 81: explain the Vannucci model.

Line 91: explain “Time pregnant rats”.

Clarify whether choosing the bregma region for injury 1.0 to -3.0 is a choice of yours or was validated previously.

Line 276: explain is the rationale for using and displaying the LC3-II/LC3-I ratio.

Figure 2: the darkness of some bands of the Western blot images, especially those of actin, may prevent accurate densitometry even with the use of a good instrument to detect Western chemiluminescence. Please discuss.

Figure 5: while the linear regression in the C panel appears reliable, that in the D panel looks improbable. Report the p or F values instead of r, and the confidence limits. I suspect that increasing n would give more statistical robustness to this elaboration.

Author Response

  1. The number of animals used, despite statistical significancy of some reported data, is not high. I understand that meeting the Three R’s and economical principles is a must in much research work, but here n does never exceed 5, with some plots with n=3. In the lack of more robust numerosity, which represents a true limitation to be highlighted, this manuscript, although interesting, should thus be labelled as “preliminary”.

Response: We have repeated the experiments that produced the data for Figure 3 to now have an n = 5-6 for each group. The results that we now report are essentially unchanged except that we lost the significance for the reduction in P62 in the Tat-Beclin1 treated animals (p <0.1) while gaining statistical significance between the Sham operated controls and the Vehicle treated H-I animals. The LC3 data remained the same. Note that for the behavioral experiments depicted in Fig. 4 that an n=9 per group was used.

  1. The low contrast in immunofluorescence photographs are prevents visual appreciation of therein contained information, both at the PC screen and when printed on paper. Increase the contrast for the blue and red colors.

Response: We have increased the brightness and contrast for figure 1 so that the blue shows better.

  1. The representation of data as bar graphs mean/SEM when n is so low induces perplexity. The usage of box and whiskers or reporting individual points as in Figure 4c is a good practice for transparency to be used for all the plots. Report the ANOVA values when comparing more than two groups.

Response: We have reported the ANOVA values within the text as requested. We also have re-formatted all of the graphs so that the individual data points are evident on the bar graphs.

Minor

  1. The introduction is quite general, especially the second para (lines 52-62) that belongs to a review rather than to a research article. The relevance of the study may be better appreciated if the Authors go directly to the central issues.

Response: Our goal was to make this article accessible to those outside the field of perinatal medicine; and in particular to highlight the distinction between the treatment time-frame used here compared to the majority of the other articles published in this field, where the tertiary period of recovery is rarely targeted. We have edited the text of the 2nd paragraph within the Introduction and tried to clarify the goals of the study better within the Introduction.

  1. Line 81: explain the Vannucci model.

Response: We have provided a citation that describes the evolution of this animal model since it was originally described by Rice et al.,1981.

  1. Line 91: explain “Time pregnant rats”.

Response: we apologize for the typographical error. This has been corrected to “Timed pregnant rats”.

  1. Clarify whether choosing the Bregma region for injury 1.0 to -3.0 is a choice of yours or was validated previously.

Response: This brain region is within the territory of the middle cerebral artery which is the territory that is affected in this model. Again, we have cited the review article for the Vannucci model.

  1. Line 276: explain is the rationale for using and displaying the LC3-II/LC3-I ratio.

Response: We have clarified in the text that LC3 conversion (LC3-II to LC3-I) was used as a measure of autophagosome production.

  1. Figure 2: the darkness of some bands of the Western blot images, especially those of actin, may prevent accurate densitometry even with the use of a good instrument to detect Western chemiluminescence. Please discuss.

Response: As stated in the Methods, we used the BioRad ChemiDoc Imaging System to collect and quantify our Western Blot data. When using films to collect Western chemiluniscence data one certainly can saturate the film and, therefore, the results will not be valid results. However, the BioRad ChemiDoc system has a dynamic range over 4 orders of magnitude, therefore, the possibility of this occurring is extremely small. We have reviewed our Western data and can confirm that we had not saturated the imaging system sensors.

  1. Figure 5: while the linear regression in the C panel appears reliable, that in the D panel looks improbable. Report the p or F values instead of r, and the confidence limits. I suspect that increasing n would give more statistical robustness to this elaboration.

Response: We have provided the F values and confidence limits in the Results for all studies where ANOVAs were performed.

Reviewer 3 Report

Cells - 1053876

The authors evaluated delayed TGFB type 1 receptor ALK5 (3 days after HI) on neurological outcome. This approach of delayed therapeutic strategy is interesting, because not so studied.

This is a well-written and organized manuscript, which needs some improvements before publication.

Specific considerations:

Major criticism: All the data are presented as mean +/- SEM. The number of animals used in each experiment is only indicated in the figure legends.

As this number is very small (n=3-4 in Fig 1, 3 and 5; n=6 in Fig. 4 – but not stated in the legend), data should be presented as dots with median and interquartile range (as in Fig. 4C, see below).

Statistical analysis

For presenting continuous measures (e.g. infarct volume) in graphs, use dot plots or boxplots. Bar charts are not appropriate for continuous measures since they do not provide information about distribution of the data. Appropriate descriptive measures of the average and variability for continuous measures in tables, text, or graphs are the arithmetic mean and standard deviation if the data are sufficiently normally distributed, or the median and interquartile range [being the 25th and 75th percentile] if data are not sufficiently normally distributed, but not the standard error (SE). The presentation of the standard error as measure of variability is not correct since it is a 67% confidence coefficient for the mean, meaning that the interval mean ± standard error of the mean is a 67% confidence interval of the mean. For presenting model-based measures (e.g. from ANOVAs), please give effect estimates and 95% confidence intervals.

Other criticisms:

Introduction, line 66: apoptotic deaths? How many deaths, provide details.

M&M, line 98: provide the strain of rat.

Line 130, indicate, for the Bregma: +1.0 to -3.0 mm.

Results and Figure 3: In the legend, indicate TB1 (for Tat-Beclin 1).

Figure 4: The number of animals is small to evaluate sensorimotor outcomes. Increase this number, at least in the HI TB1 group.

Discussion: The two first paragraphs at the beginning of the discussion are a repeat of the introduction.

Line 422-423: rephrase the sentence.

Author Response

  1. All the data are presented as mean +/- SEM. The number of animals used in each experiment is only indicated in the figure legends. As this number is very small (n=3-4 in Fig 1, 3 and 5; n=6 in Fig. 4 – but not stated in the legend), data should be presented as dots with median and interquartile range (as in Fig. 4C, see below).

Response: As indicated above, we have re-formatted all of the graphs so that the individual data points are evident on the bar graphs. All of the data points are now provided so that a reader can view the distribution of the data. We have used this presentation of the data, as the data are easier for a reader to view than dots with median and interquartile range.

Other criticisms:

  1. Introduction, line 66: apoptotic deaths? How many deaths, provide details.

Response: We have clarified the text; however, the number of dead cells is proportional to the severity of the insult; therefore, even providing a range of cell deaths is not meaningful.

  1. M&M, line 98: provide the strain of rat.

Response: We thank you for pointing out this omission. We now specify that these were Wistar Rats.

  1. Line 130, indicate, for the Bregma: +1.0 to -3.0 mm.

Response: The text has been corrected.

  1. Results and Figure 3: In the legend, indicate TB1 (for Tat-Beclin 1).

Response: The abbreviation TB1 is now specified in the figure legend.

  1. Discussion: The two first paragraphs at the beginning of the discussion are a repeat of the introduction.

Response: We are a little puzzled, by this comment as we feel that these paragraphs provide new information that was not provided in the Introduction and thus serve as an entry to Discussing the data presented and their importance.

  1. Line 422-423: rephrase the sentence.

Response: We have rephrased this sentence to clarify it.

Round 2

Reviewer 2 Report

All concerns have been addressed adequately, with the following exceptions:

Fig 5D, when P>0.05, no regression lines can be traced.

"We have reported the ANOVA values within the text as requested". This concern has not been fully met. Please report clearly the ANOVA values and the results of the Tukey's test.

Author Response

  1. We have removed the regression analysis figure from Figure 5.
  2. We apologize that some of reporting of the statistical analyses were missing from our revised manuscript - some of the changes that the first author had inserted must have been lost during the revision process by another reviewing author.   We have revised our manuscript to provide the results of the ANOVA analyses within the text of the Results. During this revision, we also noticed that we had not always specified in the figure legends whether a one-way or two-way ANOVA had been performed, so we have now provided this information as well.

Reviewer 3 Report

The manuscript is very interesting and now well presented and all the comments have been taken into account.

Author Response

We are pleased to hear that the revisions that we made to the manuscript have now been found satisfactory by this reviewer.